# Numerical Investigation of Impeller-Vaned Diffuser Interaction in a Centrifugal Compressor

**Matteo Bardelli [1], Carlo Cravero [1], Martino Marini [2,*], Davide Marsano [1] and Omar Milingi [1]**

[1] Dipartimento di Ingegneria Meccanica, Energetica, Gestionale e dei Trasporti (DIME), Università di Genova, 16145 Genova, Italy; matteo.bardelli@edu.unige.it (M.B.); cravero@unige.it (C.C.); davide.marsano@edu.unige.it (D.M.); omar.milingi@gmail.com (O.M.)

[2] Dipartimento di Architettura, Design e Urbanistica (DADU), Università di Sassari, 07041 Alghero, Italy

[*] Correspondence: marini@uniss.it; Tel.: +39-079-9720-409

**Abstract:** The work presents the results of a CFD campaign to investigate the impeller–diffuser interaction in a centrifugal compressor, taking advantage of experimental data from the open literature. Previous studies on the same turbomachine focused on an experimental investigation to understand the flow interaction between the impeller and the vaned diffuser. These experimental data have been used to validate the simulation approach and discuss its results. Several CFD models with increasing complexity have been developed to take into account different aspects. The steady analysis has been performed to highlight the potentials and limitations of such models and to carry out a first study of the flow. In order to analyze the impeller–diffuser interaction, a further model for the unsteady analysis has been set up. Two different operating points have been investigated: one on the surge limit and another in a more stable working zone. A good agreement with the experimental reference data has been obtained with the unsteady analysis and some insights in the complex flow field are deduced.

**Keywords:** centrifugal compressor; CFD; impeller–diffuser interaction

## 1. Introduction

The interaction between impeller and diffuser in centrifugal compressors is characterized by complex flow phenomena, which are still the focus of several research studies. Such an interaction, characterized by a set of mechanisms which causes distortions in the flow pattern, cannot be pinpointed if the respective components are studied independently. The impeller, because of secondary flows, gives rise to an outlet flow with a strongly unsteady character; the flow coming from the impeller would be non-uniform even after an ideal circumferential averaging, due to the above secondary flows which generate strong differences of relative velocity and flow angle along the hub to shroud direction. When the effects of viscous transport phenomena are considerable, the flow at impeller exit is typically characterized with a jet and wake pattern (Dean and Senoo, 1960) [1]. As long as the mixing process takes place rapidly, it cannot be completed in the vaneless space, so that unsteady effects are more or less significant, depending on the radial gap. Despite the mixing process taking place rapidly, in the vaneless space the unsteady effects of the impeller are not damped before the diffuser blade; these effects are significant and depend on the radial gap between impeller and diffuser vane. On the other hand, the diffuser vanes cause a speed decrease to ascribe to a potential flow effect (Dean, 1971) [2], unsteady pressure disturbances, and Mach number at the rotor-stator interface (Bulot and Trébinjac, 2009) [3].

Numerical investigations provide good chances to split the different phenomena, especially through the comparison between a steady calculation with a mixing plane at impeller–diffuser interface

and a transient rotor-stator calculation (Yamane et al., 1993 [4], Dawes, 1994 [5], Sato and He, 1999 [6], Shum et al., 2000 [7], Robinson et al., 2012 [8], Younsi et al., 2017 [9]). Many numerical simulations have been developed and applied to study the flow unsteadiness in vaned diffusers (Krain, 2002) [10], in vaneless diffusers (Kalinkevych and Shcherbakov, 2013) [11], and to compare the two types (Cui, 2005) [12]. The effect of the radial gap between the impeller and vaned diffuser has been investigated also (Hosseini et al., 2017) [13]. In the last years, particular attention has been focused on the near surge conditions to better understand the stall mechanism (Bousquet et al., 2014 [14], Fujisawa and Ohta, 2017 [15]) and to devise some criteria to predict the surge limit in centrifugal compressors (Carretta et al., 2017 [16], Cravero and Marsano, 2017 [17]). The interaction between rotor and stator in a centrifugal compressor has also been extensively investigated through experimental analysis (Fisher and Inoue, 1981 [18], Rodgers, 1982 [19]).

A previous study on this centrifugal compressor was presented in 2002 by Ziegler [20], who conducted an experimental investigation to understand the flow interaction between the impeller and the vaned diffuser. In this paper, a CFD analysis of the compressor object of the Ziegler's studies is shown; the detailed geometry and experimental data are available for the test case called Radiver. The Ziegler's studies show that a better performance is obtained when the radial gap is small; in this condition, the rotor wake ingested by the diffuser channel is capable of energizing the diffuser boundary layer, preventing flow separation. For larger radial gaps the mixing processes cause a more radial wake flow in the vaneless space, reducing the above positive effect. In the following, the paper begins with an overview of the reference test case, then the CFD modelling is explained with mesh generation, pre-processing, and steady simulations. The second half of the paper is dedicated to the description of the numerical results, the comparison with experimental data and the discussion of the flow patterns, with special reference to the unsteady effects.

## 2. Validation of the CFD Approach

### 2.1. Reference Case

The reference case is a centrifugal compressor part of a test rig that was described by Justen et al. (1999) [21], Ziegler et al. (2000) [22], and Ziegler et al. (2002) [23,24]. The data of the CFD test case can be obtained on request by the Institute of Jet Propulsion and Turbomachinery of Aachen. The impeller, developed by MTU Aero Engines, is unshrouded and has 15 backswept aluminum blades. The wedged diffuser, designed at the Aachen institute, has 23 blades and allows a continuous adjustment of diffuser vane angle ($\alpha_{4SS}$) and radial gap ($r_4/r_2$). Some technical data are presented in Table 1. The measurements of the test case were carried out at 80% rotational speed (N = 28,160 rpm) and the numerical simulations were consequently conducted at that speed. Unsteady measurements were carried out for two diffuser geometries: [$\alpha_{4SS}$ = 16.5°, $r_4/r_2$ = 1.04] and [$\alpha_{4SS}$ = 16.5°, $r_4/r_2$ = 1.10]. Only the configuration [$\alpha_{4SS}$ = 16.5°, $r_4/r_2$ = 1.04] has been considered for the CFD simulations. The results that will be shown in the next sections are relative to the sections in Figure 1, so the CFD analysis can be compared to the experimental campaign taking into account such a nomenclature.

### 2.2. Mesh and Computational Domain

The computational domain has been built on the basis of the geometrical coordinates of the hub, shroud and blade provided by Ziegler (2002) [20], for both the impeller and the vaned diffuser. Structured grids have been generated with Turbogrid for the following parts, as reported in Figure 2: (A) single channel of impeller and a portion of the convergent duct, (B) single channel of vaned diffuser and a portion of the vaneless diffuser.

**Table 1.** Technical data of compressor for nominal speed and diffuser reference geometry.

| Compressor: | | |
|---|---|---|
| Rotational speed at design point | $N_0$ | 35,200 rpm |
| Maximum total pressure ratio | $\pi_{t,max}$ | 4.07 |
| Maximum isentropic efficiency (tot/tot) | $\eta_{stt,max}$ | 0.834 |
| Mass flow at maximum efficiency | $\dot{m}_{corr}$ | 1.956 kg/s |
| **Impeller:** | | |
| Exit radius | $r_2$ | 135 mm |
| Number of blades | $Z_I$ | 15 |
| Blade backsweep angle at exit | $\beta_{bs}$ | 38° |
| **Diffuser:** | | |
| Radial gap | $r_4/r_2$ | 1.10 |
| Diffuser height | $b$ | 11.1 mm |
| Number of blades | $Z_D$ | 23 |
| Vane angle | $\alpha_{4SS}$ | 16.5° |
| Vane wedge angle | $\alpha_V$ | 6.615° |

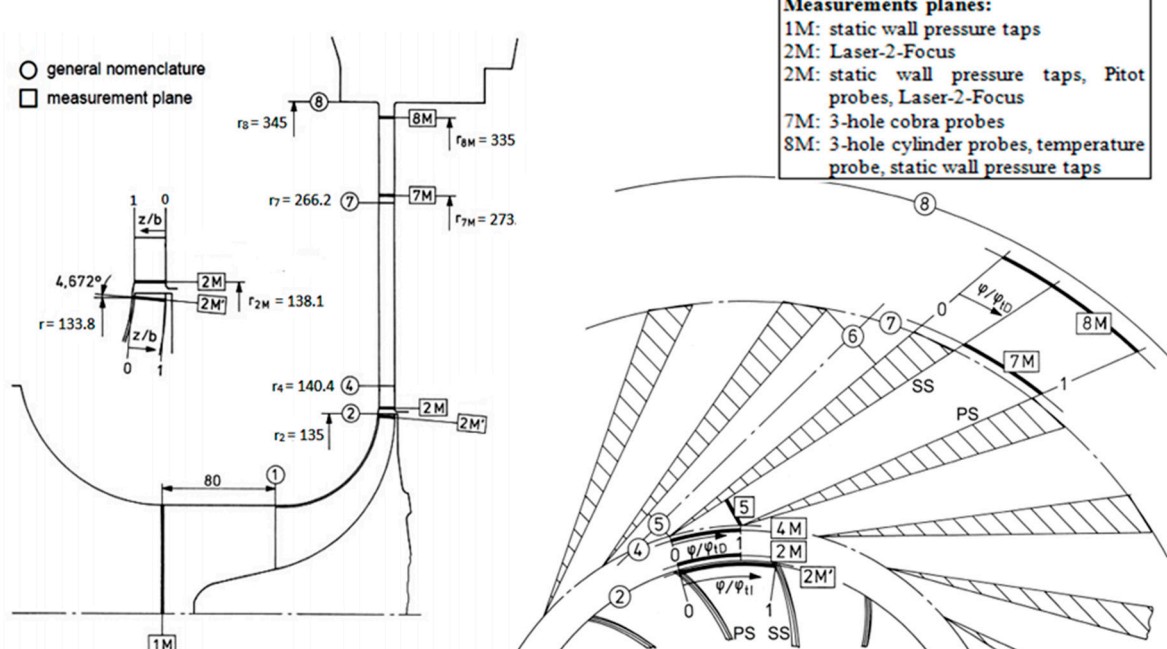

**Figure 1.** Measurement planes in meridional (left) and blade to blade (right) view (Ziegler, 2002a) [23]. Dimensions are in [mm] and referred to the configuration [$\alpha_{4SS}$ = 16.5°, $r_4/r_2$ = 1.04].

The computational domain upstream of the rotor is composed by a non-rotating subdomain from the inlet section to the rotor ogive where the rotating domain starts; at this point, an interface plane with no mixing is defined. The impeller channel is discretized with an O-grid that consists of 40 elements along the blade height. The tip clearance is subdivided into 30 mesh elements. The resulting $y^+$ value is 0.5 at the hub and 0.467 on the blade. The convergent duct is discretized with a coarser H-grid to save grid elements in a zone without complex flow structures. The diffuser channel is also discretized with an O-grid with 45 elements along the height and this yields $y^+$ = 0.5 at the hub and shroud walls and $y^+$ = 0.631 on the blade. The above values of $y^+$ are average values on the corresponding wall. The mesh of the convergent duct + impeller consists of $2.4 \times 10^6$ elements, while the mesh of the diffuser has $0.5 \times 10^6$ elements.

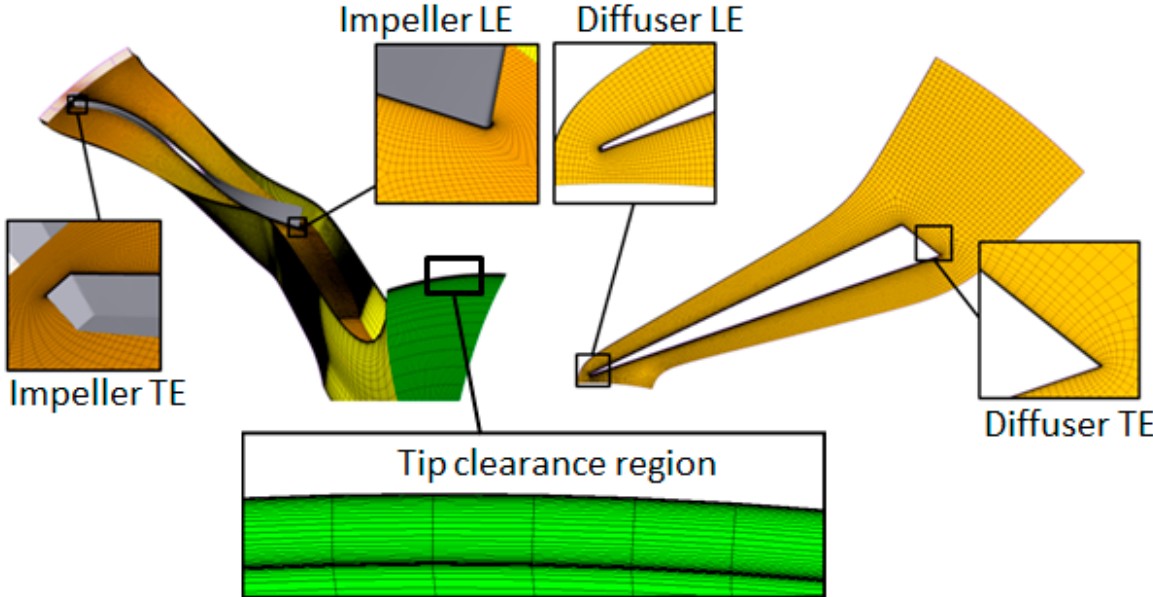

**Figure 2.** Mesh details of [convergent duct + impeller] (left) and diffuser (right).

All CFD simulations have been carried out with 2 impeller channels and 3 diffuser vanes without the volute.

This solution gives a ratio of the circumferential pitch of the rotor domain very close to the circumferential pitch of the stator domain (diffuser vanes) with a pitch ratio at the impeller–diffuser interface very close to one (PR = 1.02). With the above condition, the use of periodic flow option in the circumferential direction is also acceptable for unsteady simulations. The whole CFD model domain consists of $6.3 \times 10^6$ elements. A mesh sensitivity has been performed to check that the compressor performance undergoes a variation of less than 1% with the different meshes. The mesh sensitivity and mesh parameters have been tested according to previous experience (Carretta et al., 2017) [16].

*2.3. CFD Models*

The ANSYS CFX V17 software platform has been used for the CFD simulations. Figure 3 shows the different CFD models which have been tested: in each box the first row describes the analysis type (steady or transient) and the interface option between impeller and diffuser (Stage, i.e., mixing plane or transient), the second row denotes the advection scheme and the third row refers to the turbulence model. The CFX nomenclature is used, in particular, for the advection upwind scheme, which corresponds to a first order accuracy, while high resolution corresponds to a second order accuracy. The turbulence models adopted are the k-ω, the SST and the model k-ε with scalable wall functions. The following boundary conditions have been set: inlet total pressure and temperature ($T_{t\,In}$, $p_{t\,In}$), inlet turbulence intensity (5%), outlet mass-flow rate and a uniform impeller rotational speed. The mass flow rate at the exit boundary has been set, instead of the usual static pressure, because the analyses have been performed far from the choking condition. Moreover, a circumferential periodicity has been fixed because the model includes only a sector of the compressor, consisting of two impeller channels and three diffuser channels. Finally, all the solid walls have been modelled as adiabatic with no slip condition.

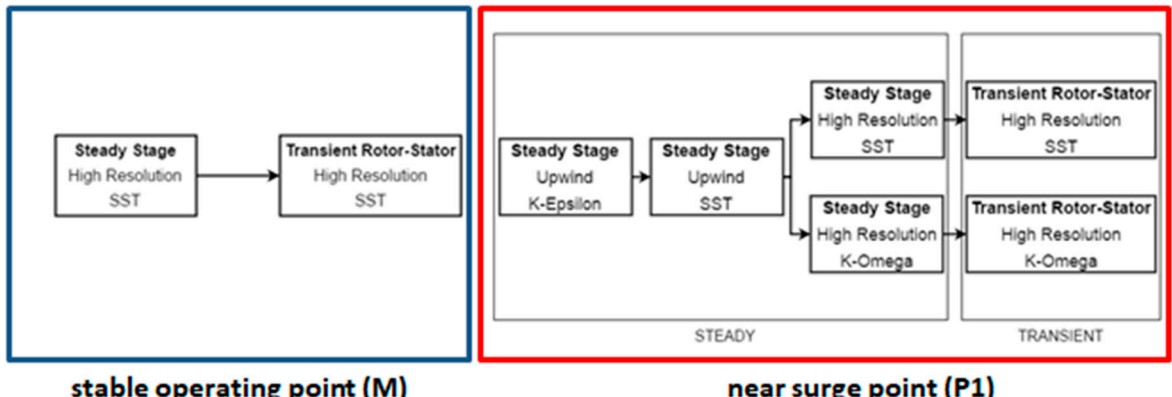

**Figure 3.** CFD simulation dataset for operating point M (close to design) and P1 (close to surge).

Two operating conditions have been simulated: point M (close to design) and point P1 (close to surge). The numerical simulations for the operating point M did not show stability problems. On the other hand, point P1 has been investigated with different models because of numerical instabilities in the original CFD settings. The activation of the turbulence model k-ω turned out to be necessary in order to dampen the instabilities. In Figure 3 a sketch of the simulation dataset for both operating points is shown.

Steady models have been validated after comparing the area averaged numerical results with the corresponding experimental results along the measurement planes, 2M and 7M, as suggested by Ziegler [24]. In all the simulations, a maximum error lower than 5% for the main averaged flow quantities (p, $p_t$, $T_t$, Ma, α) in the above sections has been obtained with respect to the experimental data. The only exception concerns the results with the model HR-SST in the operating point P1, where a stall cell appeared in one of the three diffuser channels and errors for Mach number and flow angle are much greater. The simulation interested by the stall has one low-velocity channel and two channels with high velocity (because of redistribution of mass flow), as shown by the contours of Mach number in Figure 4. No stall is calculated if, in the operating point P1, a turbulence model k-ω is used. The stall simulated in P1 with the model HR-SST induces an additional energy loss that is the cause of a lower pressure ratio at the diffuser exit, as shown in Figure 5, where the predicted overall performance is compared with experimental data. However, it must be considered that the stall in a three-diffuser vane model has a much greater relative effect on the performance than it would have in a complete model. This occurs because the mass flow redistributes over three channels rather than over the whole annulus, giving a more relevant change in incidence angle to the blade channels involved.

Transient simulations have been validated by checking that both area-averaged and local results had their time-averaged values close to the corresponding values from steady analysis. Several monitoring points have been defined at the same probe positions of the measurement devices. The stall by adopting the turbulence model HR-SST in P1 is still present in the transient calculation and is confirmed inside the same diffuser vane. The reason is due to the inability of a partial model to simulate an unsteady circumferential phenomenon: a circumferential periodicity is imposed, which forces the physical phenomenon (as previously observed, the mass flow rate blocked by the stall is redistributed in the 2 other channels of the domain and not in the remaining 22 of the complete diffuser). In all the unsteady simulations, a time step of $8.87 \times 10^{-6}$ [s] is set, discretizing the passage by a step of 1.5°. These simulations have been carried out until a perfect periodicity of the flow phenomenon is detected. In particular, after ten complete revolutions of the impeller, the frequency observed on the signal at the impeller outlet is equal to the passage frequency.

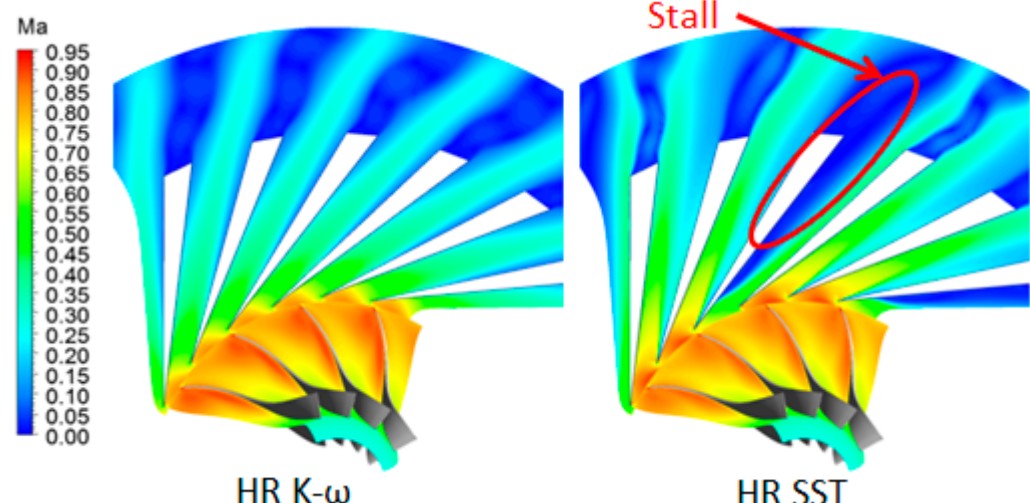

**Figure 4.** Mach number distribution at z/b = 0.5 for steady simulations HR k-ω (left) and HR-SST (right) in the operating point P1.

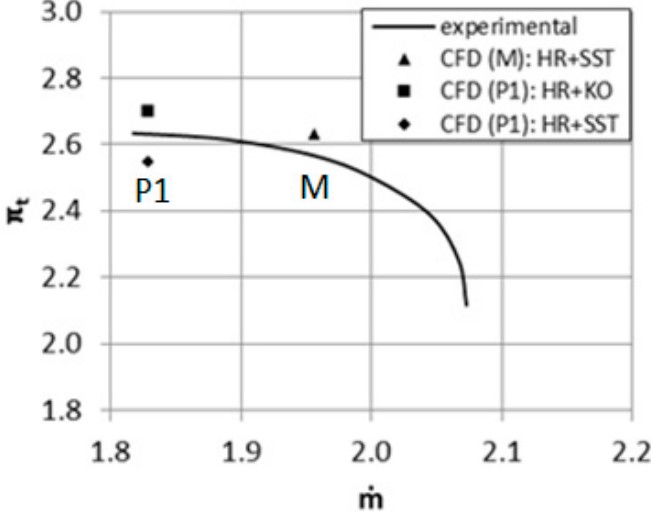

**Figure 5.** Total pressure ratio at plane 8M.

## 3. Analysis of the Impeller–Diffuser Interaction

### 3.1. Steady Analysis at 2M

The time-averaged quantities were compared to the corresponding values for the steady analysis and the experimental data. Figures 6 and 7 show the pitchwise and spanwise distributions of $p_t$ for the operating points M and P1. The numerical and experimental data show the same trend, with a tendency of the numerical results to underestimate the total pressure values; the difference is comparatively small and reaches 10.9% at its maximum. In the case of P1 point (Figure 7), the total pressure is well-captured using the k-ω turbulence model for up to 50% of the span. The model results become less accurate moving near the diffuser front wall. In this zone (z/b = 0.7 and z/b = 0.9), characterized by a strong whirling flow, the SST simulations seem to provide a better matching with the test case reference values. The spanwise distribution of $p_t$ has a maximum at z/b = 0.7 for both the operating points M and P1, while the circumferential distribution is almost constant: potential flow effects do not influence the total pressure distribution, which is only determined by the impeller geometry and operating point (inlet conditions, rotational speed). The discussion of the z/b distributions needs a preliminary investigation on the contributions of static (p) and dynamic (dependent on Ma) components that make

up the total pressure values. Only the simulation HR-KO will be considered for the point P1, because of its higher numerical stability and better coherence with the experimental data.

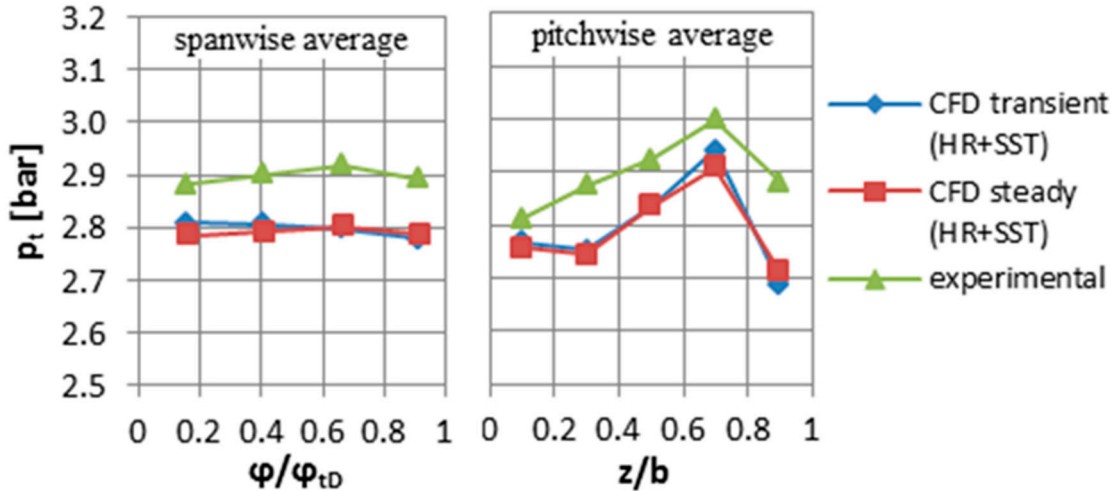

**Figure 6.** Total pressure at 2M, point M.

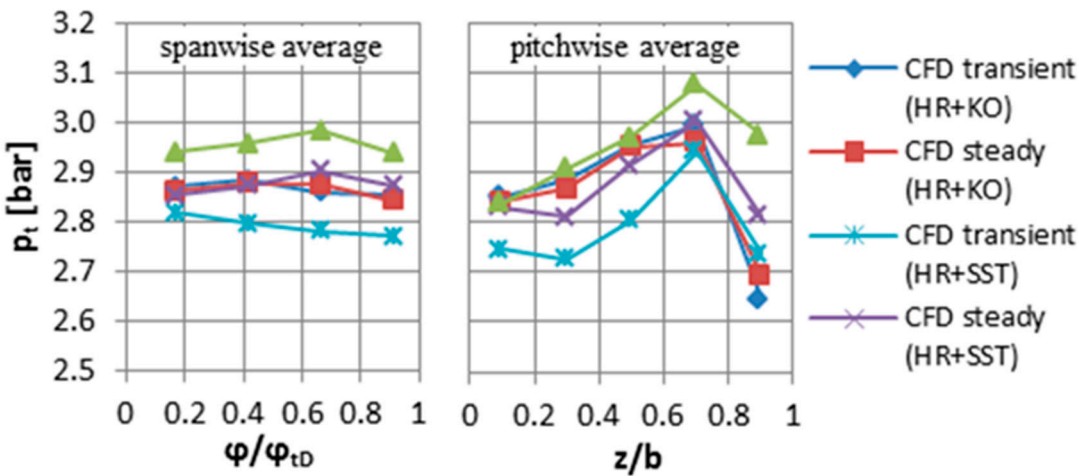

**Figure 7.** Total pressure at 2M section, point P1P1.

The maximum of $p_t$ at z/b = 0.7 is merely caused by the dynamic component, as shown by the corresponding maximum of Ma in Figure 8, because the static pressure is almost constant along z/b in 2M (z/b distribution of p in Figure 8). Because of the low meridional velocity at the surge limit, the flow is essentially tangential in 2M. Therefore, it attacks with a high incidence angle the stator blade leading edge: the stator potential flow field causes a much greater blockage (i.e., higher pressure and lower Mach number) near the diffuser PS than near the SS, as can be seen in the pitchwise distributions of p and Ma (Figure 8).

Assuming the inviscid flow hypothesis, the velocity variations in the meridional and blade-to-blade planes can be calculated separately, as first proposed by Wu (1952) [25]. Neglecting the force due to the blade lean angle ($\frac{\partial \beta_{\text{blade}}}{\partial n} = 0$), the hub to shroud velocity variation in the meridional plane is given by the following relation, which has been applied at the impeller exit (n = z, $R_n \rightarrow \infty$, $\gamma = 90°$, u = constant):

$$\frac{\partial w}{\partial n} = \frac{w \cdot \sin^2 \beta}{R_n} - \cos \gamma \cdot \cos \beta \left( \frac{w \cdot \cos \beta}{r} - 2\Omega \right) \Rightarrow \frac{\partial w}{\partial z} = \frac{\partial c}{\partial z} = 0 \tag{1}$$

where $n$ is the coordinate perpendicular to the axisymmetric streamsurface, $R_n$ is the curvature radius of the meridional streamline and $\gamma$ is the angle between streamline and axial direction in the meridional plane.

Equation (1) suggests that, under the inviscid flow hypothesis, the hub to shroud velocity distribution is constant at the impeller exit. Therefore, the variable distribution of Ma along z/b shown in Figure 8 is due to viscous phenomena inside the impeller.

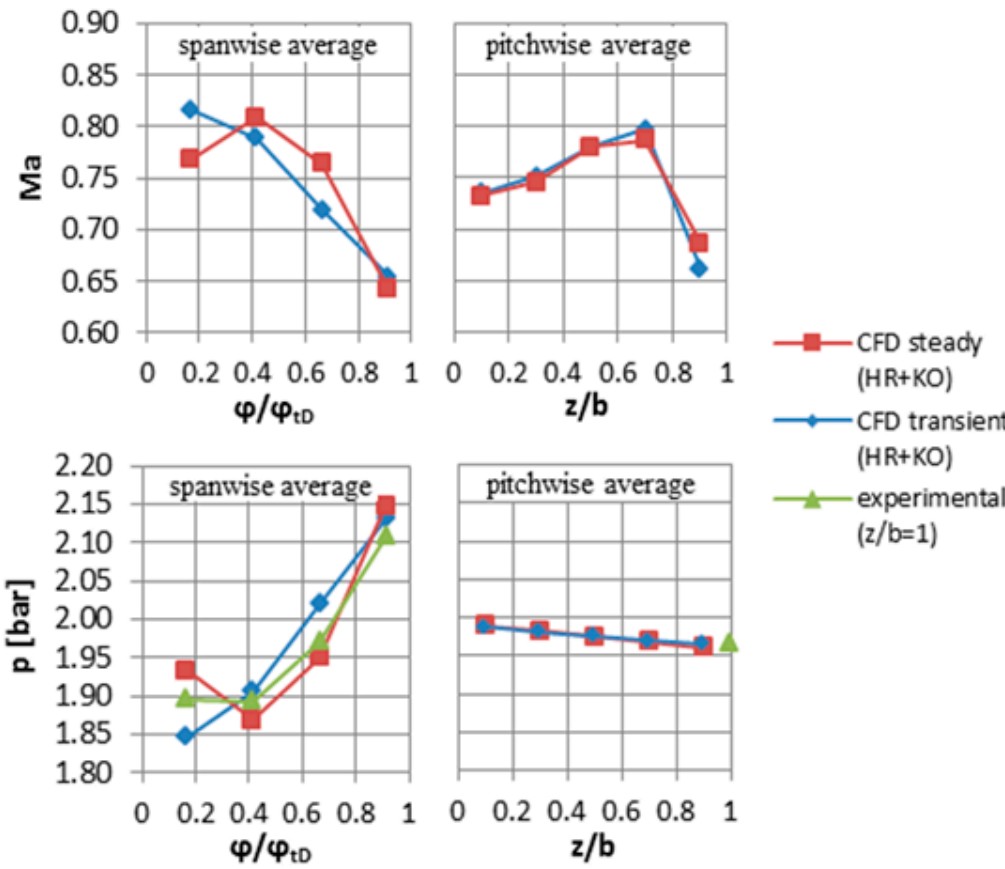

**Figure 8.** Mach number and pressure at 2M and P1 point.

*3.2. Impeller Flow Analysis*

Figure 9 shows the meridional velocity contours at five hub-to-shroud sections of the impeller and Figure 10 presents the corresponding turbulent kinetic energy contours. At surface 1, the meridional velocity is clearly influenced by blade loading (with a faster relative flow at SS) and by different curvatures of the walls (which are the cause for the hub to shroud gradient). The meridional velocity distribution is progressively distorted downstream by the increasing thickness of the shroud boundary layer; this can be seen at surface 3 of Figure 11. The growth of this low momentum region corresponds to the increase of the high turbulence zone at the casing shown in Figure 10. A better knowledge of the complex phenomena inside the rotor can be achieved through the analysis of the vorticity contours of Figure 11. The different contributions to secondary flows can be seen clearly in surface 1: two blade surface vortices along the blade height (at SS: BVS, at PS: BVP, i.e., blade surface vortices at the respective sides) and a strong vortex near the casing. This last vortex results from the merging of the passage vortex at shroud (PVS) and the Coriolis vortex (CV), as discussed by Van den Braembussche [26]. The blade vortex is stronger at the SS, due to the greater velocity gradient of the boundary layer, and tends to vanish towards the radial exit, as shown in Figure 11. This is consistent with the conservation equation of the vorticity along a streamline derived by Smith [27] and developed by Hawthorne [28]. The blade vorticity carries low energy fluid along the blade, from the hub to the

tip. PVS (passage vortex at shroud) and CV (Coriolis vortex) contribute to the transport of low energy fluid from PS to SS along the shroud wall. The passage vortex is typically stronger in the first half of the impeller (because of higher blade-to-blade curvature) and at the shroud, while in the radial portion, the Coriolis force prevails (Kang and Hirsch, 2001) [29]; in this case, the effect at the hub is not clearly visible. The Coriolis vortex (CV) and the passage vortex (PV) have the same effect and are not always distinguishable one from the other: they are often referred to as passage vortices. The overall effect is a low energy fluid transport towards the shroud wall and the SS. Moreover, at the casing, the above secondary flows interact with the tip vortex (TV). The passage vortex is well-defined in the first part of the impeller (Figure 11, surface 1) but it is clearly distorted by the trailing vortex downward and tends to move towards midspan. Weiss [30] described this phenomenon for the same compressor. The overall result is a thick high-turbulence region in the upper half of the channel (shown in Figure 10 for surface 5) that corresponds to the low relative velocity region in Figure 12 (wake). The remaining portion of the section has low turbulence level with high flow velocity and identifies the jet zone in the jet-wake model.

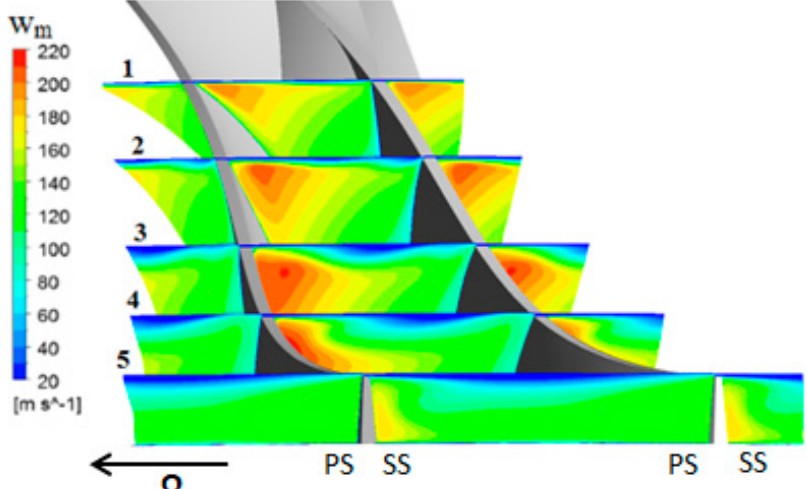

**Figure 9.** Meridional velocity in the impeller, point P1.

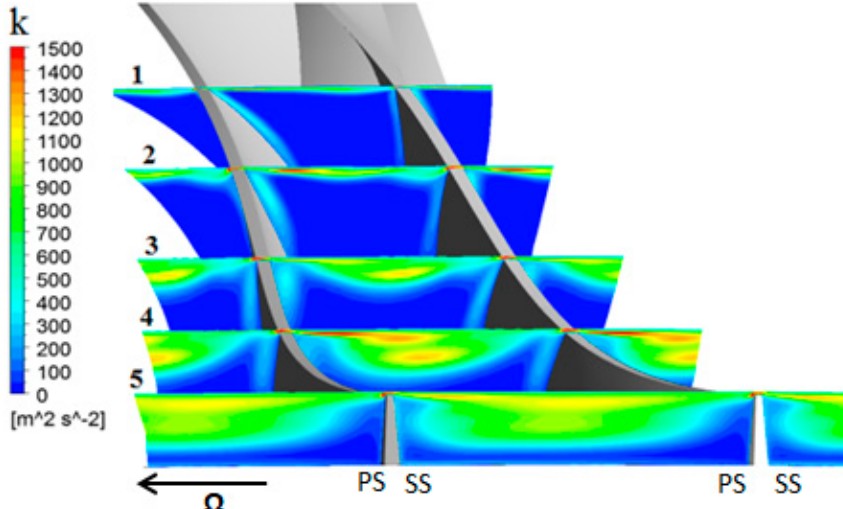

**Figure 10.** Turbulence kinetic energy in the impeller, point P1.

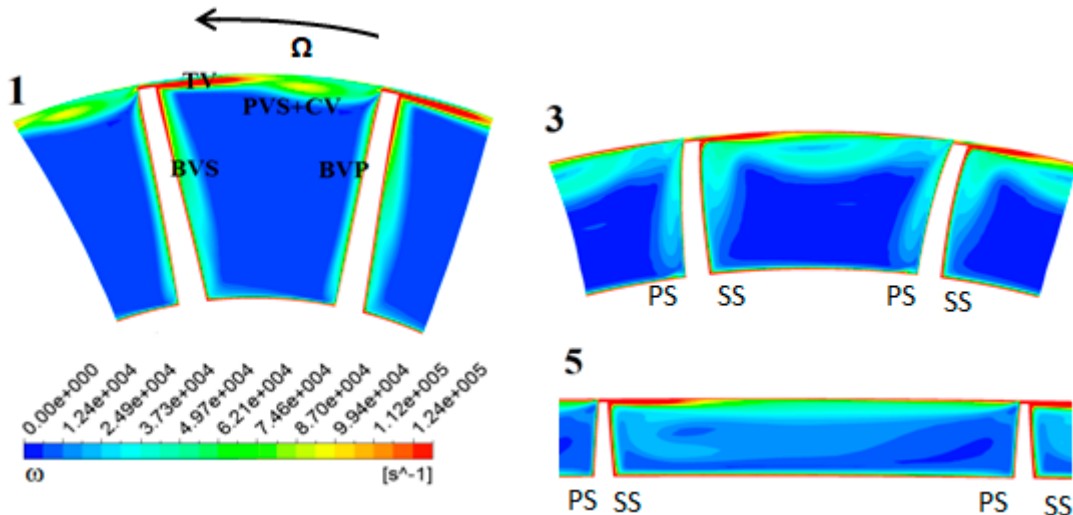

**Figure 11.** Vorticity in three hub-to-shroud surfaces of the impeller, point P1. PVS—passage vortex at shroud; BVS and BVP—blade surface vortices at SS and PS; CV—Coriolis vortex.

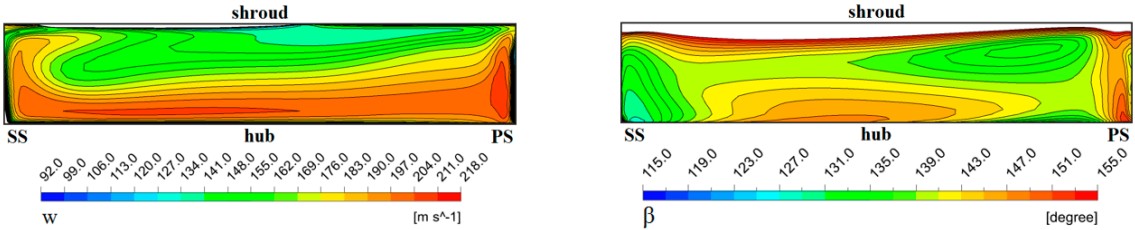

**Figure 12.** Relative velocity and angle at impeller exit in the relative frame of reference, point P1.

The velocity contours in the relative frame are shown in Figure 12 in a plane close to the mixing-plane interface on the impeller side. On the same plane, the relative flow angle distribution shows a thin stripe near the casing; here, the relative flow is completely tangential because of the tip leakage vortex and the fixed (counter-rotating in the relative frame) shroud wall. Close to the blades, the relative angle becomes more and more affected by the trailing edge local distortion. The relative flow near PS is more tangential, while the relative flow near SS is more radial and they join downstream the trailing edge. Towards the channel midspan, the flow in the wake zone has a more radial character than in the jet zone, due to the low energy transport from pressure side to suction side caused by the passage vortex.

The flow structure in the absolute frame can be discussed with the aid of the local velocity triangle sketched in Figure 13. High absolute velocities and tangential flow direction characterize the wake zone (neglecting the flow distortion at the endwalls) while low absolute velocities and radial flow direction are encountered in the jet zone, as shown in Figure 14. The relative angle distortion close to the blades is the reason for lower PS and higher SS absolute velocities. The absolute velocity maximum value occurs approximately at 70% of the channel height. The above considerations explain the Mach number distribution along z/b direction presented in Figure 8 and the total pressure curves of Figure 7. A very similar flow structure is observed for the operating point M (Milingi, 2016) [31].

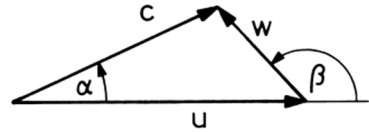

**Figure 13.** Velocity triangle.

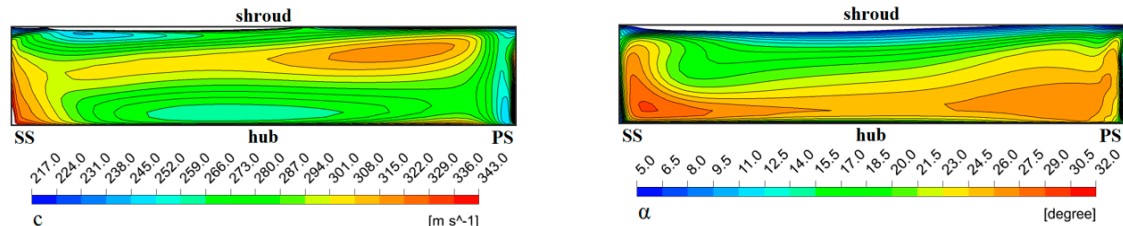

**Figure 14.** Absolute velocity and angle at impeller exit in the relative frame of reference, point P1.

### 3.3. Unsteady Analysis at 2M

In this section, the results from the transient analysis are discussed to understand the effect of the non-uniform rotating flow pattern coming from the impeller on the vaned diffuser. The analysis is focused on plane 2M for operating point P1 to get a direct comparison with the experimental data. Because of random sampling inherent to the L2F technique and due to the need for a constant time-step in the simulations, there is a minor shift between numerical and experimental instants (with absolute errors between 0.1% and 2.6%). The analysis is performed using the same rotor-stator positions from measurements. Figures 15 and 16 are diffuser-sided views in the absolute frame of reference: the impeller moves from right to left. The numerical results have been obtained from the transient simulation approach HR-KO discussed in the previous section. A good matching between the numerical and the experimental data can be observed with very similar flow patterns. The main difference takes place in the velocity distribution at the unsteady position $\varphi_{ID}/\varphi_{tI} = 0.595$. At that instant, experimental results show a high velocity zone extending clearly from right to left along the front wall and spreading towards the rear wall; this stretching is not completely detected with the numerical simulation. On the other hand, a velocity pattern similar to the experimental data has been obtained for the two transient simulations using the SST model for the operating points M and P1. Nevertheless, in the operating point P1, the results with SST are shifted because of the stall appearing downstream in the diffuser channel, so consequently the flow is slower and more tangential at plane 2M. The results using the SST model are not used for further discussion because of the instability produced by the stalled channel in the diffuser, as previously discussed.

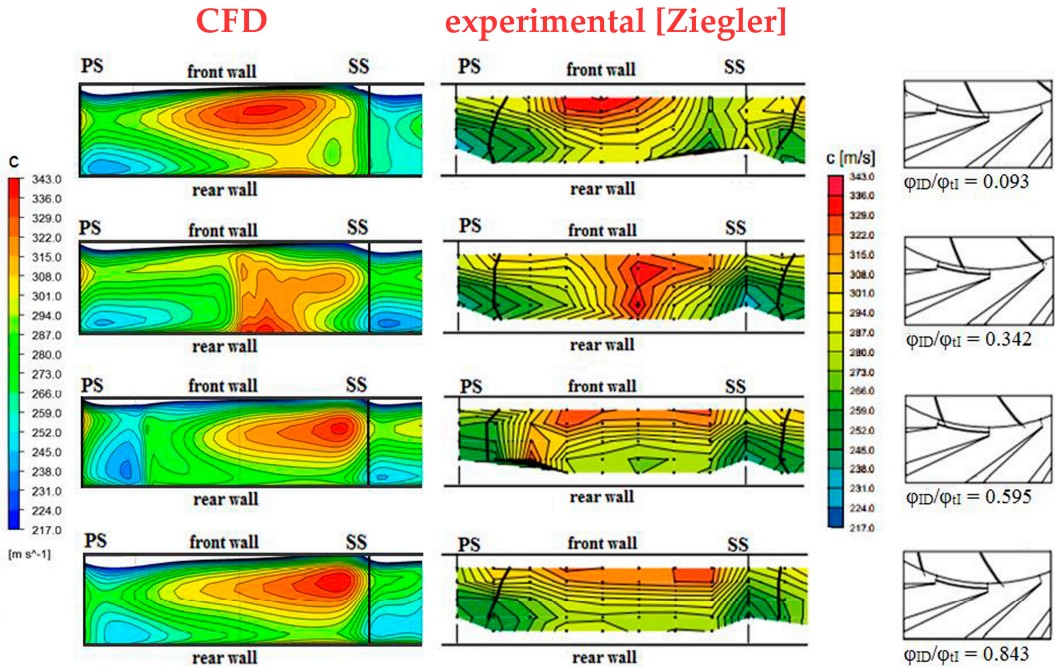

**Figure 15.** Unsteady absolute velocity at 2M in the absolute frame of reference, point P1, comparison between numerical results of HR-KO (left) and experimental results (right).

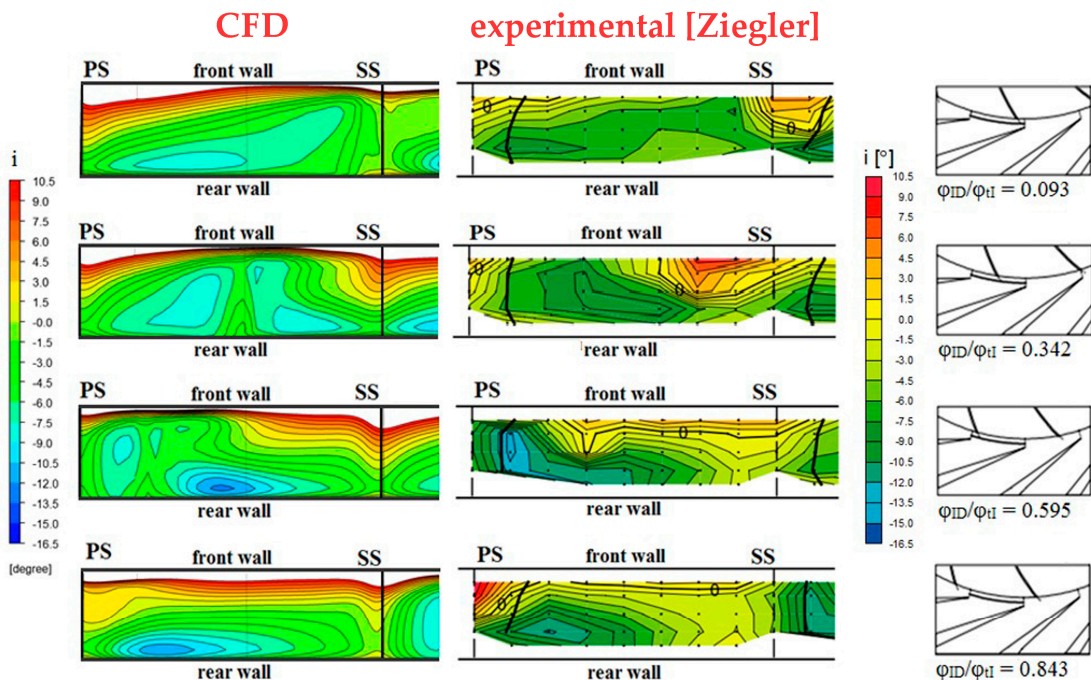

**Figure 16.** Unsteady incidence at 2M in the absolute frame of reference, point P1, comparison between numerical results of HR-KO (left) and experimental results (right).

Many of the effects studied separately in the previous sections using steady analysis can now be seen in the single unsteady analysis. Flow incidence at the diffuser is positive (i.e., absolute flow is more tangential with respect to the diffuser vane camber line) almost exclusively near the front wall where the wake generates a highly tangential absolute flow. On the other hand, the incidence angle becomes lower (and negative) towards the rear wall (jet zone). Furthermore, a thicker positive-incidence zone can be seen from the right to the left in the center of the wake zone moving with the impeller channel.

The flow has a higher velocity close to the front wall because of the jet-wake flow pattern. The absolute velocity maximum values are located at about $z/b = 70\%$ as shown by the monitoring points and by the vortical structures analysis. The absolute velocity becomes lower and the flow more radial (negative incidence) at the diffuser vane leading edge on the right side because of the diffuser vane blockage; the opposite occurs on the left side, with respect to the vane leading edge. A velocity maximum can therefore be seen close to the front wall on the diffuser vane suction side, while a minimum is found at the opposite corner (between the rear wall and the pressure side). The velocity peak is more evident when the high-velocity zones in the impeller flow (Figure 13) are aligned with those caused by the potential flow field: this occurs at $\varphi_{ID}/\varphi_{tI} = 0.093$.

A zone with high negative incidence values appears along the rear wall towards the vane diffuser pressure side (see instants corresponding to $\varphi_{ID}/\varphi_{tI} = 0.093 - 0.843$). It is clearly distorted by the passage of the impeller blade trailing edge (see instants $\varphi_{ID}/\varphi_{tI} = 0.342 - 0.595$) with high flow incidence.

Figure 17 displays the pressure fluctuation in a monitoring point placed close to the center of plane 2M, for the three transient calculations.

Two types of fluctuations are visible: a former with a high frequency and a latter with a low frequency, which corresponds to three times the impeller rotational frequency. The former is caused by the impeller blade passage while the latter is probably due to downstream disturbances (e.g., diffuser vane edges). The simulation for the operating point P1 with the HR-KO model shows lower fluctuations compared to the results with the HR-SST model. In the case of operating point M, the HR-SST model shows similar high frequency oscillations but greater low frequency oscillations. Therefore, the SST tends to generate stronger downstream disturbances, as confirmed by the P1 with HR-SST case where high frequency oscillations are also much more evident. This is assessed as the reason for the stall

detection at the operating point P1 using the SST model. The stall would be a rotating stall, passing to the neighboring diffuser vanes, but with the 3-vane periodic model, the simulation of the phenomenon is inhibited.

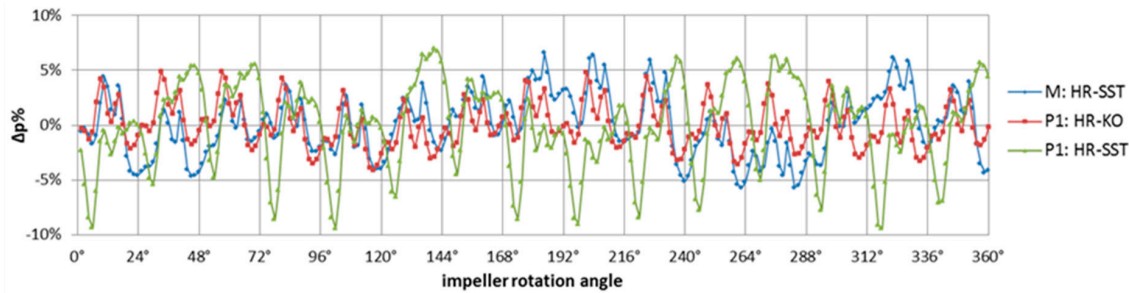

**Figure 17.** Pressure fluctuation in percentage at 2M for $\varphi/\varphi_{tD} = 0.4145$ (z/b = 0.5).

## 4. Conclusions

The role and features of different CFD models to investigate the flow in the centrifugal stage configuration of a dynamic compressor have been discussed; a quite detailed comparison between the numerical results obtained and corresponding experimental data has been possible thanks to the rich dataset from Ziegler et al. (2002) [23,24]. The simplified approach with a reduced number of channels has been discussed with respect to the full annulus analysis. The steady simulations have allowed by a first analysis to identify the main phenomena occurring at the impeller-vaned diffuser interface. Diffuser potential flow field causes a lower absolute velocity at the stator leading-edge projection on plane 2M. At the operating point close to surge, the velocity deficit at the impeller outlet plane becomes more evident and moves towards the diffuser pressure side because of a more tangential flow. A study of vortical structures in the rotor has helped to understand the impeller flow structure. A strong passage vortex, transferring low energy flow from pressure to suction side in the impeller channel, interacts with the opposite tangential flow due to the tip vortex. The final effect is an evident jet-wake flow pattern at the impeller exit with the wake zone placed in the upper half along the channel height (close to shroud) and with the jet zone in the remaining part. A good agreement with the experimental data has been obtained, with the unsteady analysis of impeller–diffuser interaction and the instantaneous data giving a further insight into the flow structure at the impeller exit. A peak with high velocity value is detected close to the front wall and SS diffuser vane with the lowest value placed at the opposite corner (rear wall and PS projection). The SST turbulence model has identified a stall cell in the diffuser channel that was not detected by the k-$\omega$ model and no clear evidence of this is shown in the experimental data. It could be a rotating stall, not highlighted by the experiments, that cannot be simulated with a periodic 3-vane model. The simulations in the operating point P1 using the k-$\omega$ model show a good agreement with experimental data.

**Author Contributions:** The contribution of all the authors has to be subdivided on to an equal basis. M.M. and C.C. designed the methodology and revised the draft; M.B., D.M. and O.M. contributed to calculations in different phases and interpretation of results.

**Funding:** This research received no external specific funding.

**Acknowledgments:** The above analysis has been possible thanks to the detailed data set (Radiver test case) provided by Niehuis. The authors are grateful to Niehuis, Ziegler, and Institute of Jet Propulsion and Turbomachinery for the sharing of their precious results and data.

**Conflicts of Interest:** The authors declare no conflict of interest

## Nomenclature

| | |
|---|---|
| c | absolute velocity |
| $c_m$ | meridional component of c |
| i | incidence angle to the diffuser camber line |
| k | turbulent kinetic energy |
| $\dot{m}$ | mass flow rate |
| Ma | Mach number |
| N | shaft speed |
| p | pressure |
| r | radius |
| T | temperature |
| u | circumferential speed ($\Omega_r$) |
| w | relative velocity |
| z/b | relative coordinate normal to the diffuser rear wall (Figure 1) |
| Z | number of blades |
| $\alpha$ | absolute flow angle (Figure 17) |
| $\beta$ | relative flow angle (Figure 17) |
| $\varphi/\varphi_{tD}$ | circumferential coordinate relative to the diffuser pitch (Figure 1) |
| $\varphi_{ID}/\varphi_{tI}$ | impeller unsteady position relative to the impeller pitch (see $\varphi/\varphi_{tI}$ in Figure 1) |
| $\pi$ | pressure ratio |
| $\omega$ | vorticity |
| $\Omega$ | angular velocity |

Subscripts

| | |
|---|---|
| D | Diffuser |
| In | Inlet |
| I | Impeller |
| max | maximum |
| t | total |

Acronyms

| | |
|---|---|
| HR | High Resolution |
| KE | K-Epsilon (k-$\varepsilon$) |
| KO | K-Omega (k-$\omega$) |
| PS | Pressure Side |
| SS | Suction Side |
| SST | Shear Stress Transport |
| UW | Upwind |

Operating Points

| | |
|---|---|
| M | stable (Figure 5) |
| P1 | near surge (Figure 5) |

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
