# Peer review of "Numerical Investigation of Impeller-Vaned Diffuser Interaction in a Centrifugal Compressor"

_applsci, doi:10.3390/app9081619_

Round 1

Reviewer 1 Report

The present manuscript depicts CFD results on the interaction between impeller and vaned diffuser of centrifugal compressor. The reviewer points out some important issues which should be considered by the authors:

1) In-depth literature survey on the interaction between impeller and vaned diffuser, especially in CFD point of view, should be given, together with more detailed explanations of the present CFD method.

2) Carefull validation of the present CFD should be made.

3) Transient CFD would be essential in exploring the interactions which can be specified with unsteady pressure flucations.

4)  Detailed physical explanations should be given on the resutls obtained.

Author Response

A file with a list of answers to reviewers' observations is enclosed.

Reviewer 2 Report

Summary of the Work

The paper performed the numerical investigation of impeller-vaned diffuser interaction using commercial software CFX. Both steady and unsteady simulations were performed. The simulations were performed using an existing geometry (Ziegler’s compressor). The numerical results were compared to available experimental data and pretty good agreement were achieved when using k-ω turbulence model.

Quality and Originality

The quality of the work is average. Numerical simulation of impeller-diffuser interactions has been studied many times. This work just adds another example. The numerical simulation is performed using commercial CFD software CFX, which is quite standard practice. Although decent agreement between simulation and experiments were achieved using k-ω turbulence model, the simulation using SST turbulence model was quite different from the experiments. Additionally, the simulation was performed at one r4/r2 ratio (r4/r2 =1.04), which significantly undermine the value of the work. Above all, the quality of the work is average and its long-term impact is quite limited.

Reviewer’s Recommendation

This paper will be much better if the authors include simulation results at another radius ratio (r4/r2=1.10). Hence the radial gap is a very important parameter for studies in impeller-diffuser interactions, this will open many opportunities to examine the capability of CFD in characterizing impeller-diffuser interactions.

The reviewer would recommend major revision for the manuscript.

Specific Comments

The detailed comments on the formatting and writing of the manuscript could be found in the draft attached.

Author Response

A file with the answers to reviewers' osbservations is enclosed

Reviewer 3 Report

The authors present a CFD simulation campaign aimed to investigate the impeller-diffuser interaction and to compare the results to experimental data found in literature (Ziegler).

In the introduction the authors affirm that some of the numerical work by Ziegler will be presented. In the text and in the paragraph structure is not clearly defined which part is the case study.

The presentation of the results in the figures is rather chaotic and sometimes difficult to understand.

A thorough language review is strongly recommended .

1. Introduction

The introduction is concise clear in the concept but the language should be strongly improved.

24. "This interactio, wich can be defined as..." the concept is clear but the sentence should be rewritten.

26. "Impeller..." please review the sentence.

28. "as well" do the authors mean "as well as"?

31. "Even if the mixing process.." please explain the concept in a clearer form.

49. performances

2. Validation of the CFD approach

The title is quite confusing. It is not really clear where the numerical data presented are part of the authors' simulation and where they are taken from the case study.

The steady state model requires further explanations.

Table 1. The content in technical data and reference geometry are sufficient, but it is difficult to read the table, please improve.

74. The authors speak about two different meshes and then present two different geometries. Given that the mesh will be different for the two cases, for the sake of clarity the paragraph should be reformulated and the reference to a geometry more than to a mesh would be clearer for the reader. 

Figure 2. The computational domain here used is taken from Ziegler? If so that should be underlined better in the text. Furthermore, if the figure is taken from literature this should be indicated in the caption. If figure 2 refers to the authors' work, this should be presented in a clearer way.

97. Did the author perfom a mesh sensitivity test? If so this should be mentioned, if not, can the authors explain why?

106. The authors should explain in a more extensive way the choice of boundary conditions.

Figure 3. Which k-epsilon model was used? And can the authors explain their choice?

138. Can the authors explain the convergence criteria used?

Figure 5. It should be improved

3. Analysis of the impeller-diffuser interaction.

Figures 6-7. The time averaged quantities for the steady and transient simulations are here presented alongside the experimental data. It is not possible to spot the difference between the steady and the transient in some points of the graph, this should be improved. Figure 7 misses the point in the caption.

   The model underestimates the pressure, can the authors explain if this was expected and why?

180. equation (1) is not readable in the pdf.

Figure 8 is difficult to read.

193 "downstream in the streamwise direction" please reformulate.

Figures 15-16. Numerical and experimental data are presented in a chaotic figure. If the figure regarding the experimental data is taken directly from Ziegler, this should be clearly explained. In any case if the contour have different characteristics, the use of two different figures is recommended.

297. "a former...and a letter." not clear. Please rewrite all the text in lines 297 and 298.

4. Conclusions

314-315 please rewrite

Author Response

A file with the answers to reviewers' observations is enclosed

Round 2

Reviewer 2 Report

This paper is average. Tons of CFD were performed but really nothing new was concluded from the study. In addition, only one radial gap case was investigated. The reviewer is not clear about what motivates the authors to include the reference 7,9, 17, and 18, none of those are archival papers, all conference papers. This does not improve the quality of the paper at all. 

Author Response

Enclosed the answers to reviewer's remarks

Reviewer 3 Report

Dear Authors,

thank for providing a reviewed version of the paper. My comments were addressed except for the comment regarding the convergency criteron. This is not what I asked for.

Further editing of English language is still required            

Author Response

Enclosed the rebuttal for this second phase to the observations of the reviewer
